# Characteristics of Grape Shelf Eye Injuries at Vineyards in Japan

**DOI:** 10.3390/jcm11237079

**Published:** 2022-11-29

**Authors:** Wataru Kikushima, Yoichi Sakurada, Kenji Kashiwagi

**Affiliations:** Department of Ophthalmology, University of Yamanashi, Shimokato 1110, Chuo 409-3898, Japan

**Keywords:** farm work-associated eye injury, grape breeding, open-globe injury, ophthalmic surgery, ocular trauma score

## Abstract

We aimed to investigate the characteristics and visual outcomes of farm work-associated eye injuries at vineyards. We retrospectively reviewed medical charts of patients with farm work-associated eye injuries. The eyes were divided into two groups according to the type of farming that contributed to the eye injury: the vineyard and other farming groups. Injury types, surgical procedures, and changes in visual acuity were statistically evaluated. After initial treatment, patients were followed up at different periods. We examined 30 eyes, including 14 eye injuries in the vineyard group and 16 eye injuries in the other farming group. The mean age of the patients was 58.8 ± 16.7 years, and 83.3% were male. None of the patients wore any safety eyewear at the time of injury. After initial treatment, the mean best-corrected visual acuity significantly improved from 0.83 ± 0.94 at baseline to 0.30 ± 0.57 at the final follow-up (*p* = 5.8 × 10^−4^). Eye injuries in the vineyard group were mostly caused by the penetration of wires of grape shelves and were frequent from winter to spring. We concluded that farm work-associated eye injuries at vineyards have characteristic properties compared with those during other farm work. The use of safety eyewear is strongly recommended to prevent eye injuries during farm work.

## 1. Introduction

Farming is one of the most hazardous occupations and possibly causes numerous types of traumas and/or infections [1,2]. Among farm work-associated hazards, eye injuries frequently cause fatal damage to farmers’ vision [3,4,5]. For instance, long-term exposure to sunlight and ultraviolet rays causes severe cataracts, pterygium, or corneal erosion. Contusion of the globe owing to branches, stalks, thorns, or vines occasionally encompasses traumatic hyphema, retinal detachment, or even severe open-globe rupture [6]. Furthermore, some electrical equipment such as wire brushes and grass trimmers cause metallic intraocular foreign bodies [7,8,9].

Yamanashi Prefecture is a region in Japan where fruit farming, particularly grape breeding, is thriving [10,11]. Farmers engaged in grape breeding in vineyards have a risk for farm work-associated eye injuries due to grape branches, vines, wires, or electrical equipment. However, few studies have evaluated the occupational risk of farm workers in vineyards [12].

Thus, this study aimed to investigate the detailed course of vineyard-related eye injury and reduce farmers’ risk of vision loss.

## 2. Materials and Methods

### 2.1. Participants

We retrospectively reviewed the medical charts of patients who visited the tertiary referral suite of the ophthalmology department at Yamanashi University Hospital because of farm work-associated eye injuries between April 2009 and March 2022. This study was approved by the Institutional Review Board of the University of Yamanashi and was conducted according to the tenets of the Declaration of Helsinki. On initial presentation, all patients underwent essential ophthalmic examinations, including best-corrected visual acuity (BCVA), intraocular pressure, slit-ramp biomicroscopy, and ophthalmoscopy with pupil dilation if feasible. Patients consecutively underwent emergency ophthalmic surgery, if needed. After initial treatment, patients were followed up at the outpatient clinic of the ophthalmology department during various periods (range, 1–122 months). Age, sex, baseline/final BCVA, type of eye injury, presence of infection, wearing or not wearing eye protection, and time to hospital presentation were recorded. The dates of the initial presentation were recorded quarterly.

The ocular trauma score (OTS) of the eyes included in this study was calculated based on initial BCVA and the type of injury. OTS was categorized according to the OTS study; in detail, the raw points of each patient were calculated according to baseline BCVA and the type of injury and were then converted into corresponding OTS scores (category 1: severest to category 5: mildest) [13].

The inclusion criteria were as follows: eyes injured during any farm work and those in the acute phase (within 1 week from the day of injury). The exclusion criteria were as follows: (1) eyes injured during any other activity (industrial work, sports, quarrels, falls, or traffic accidents); (2) eyes injured more than 1 week before (subacute to chronic phase); (3) eyes with a preexisting ocular disease that affects visual acuity (VA); or (4) eyes that missed their first follow up.

### 2.2. Treatment

On presentation, patients underwent emergency ophthalmic surgery if the injured eye showed corneal or scleral penetration, corneal or anterior chamber foreign body, traumatic cataract, traumatic retinal detachment, infectious endophthalmitis, lacrimal canaliculi rupture, or open-globe rupture. Although most surgeries were performed under topical anesthesia, general anesthesia was chosen in the case of open-globe rupture. Topical and general antibacterial agents were administered after initial surgery to treat or prevent infections. Tetanus vaccination was applied if the contamination of the wound was suspected.

### 2.3. Statistical Analysis

Statistical analyses were performed using StatFlex ver. 7.0 (Artec Co., Ltd., Osaka, Japan). Decimal BCVA measured using the Landolt chart was converted to the logarithm of the minimal angle resolution (logMAR) for analyses. Based on previous studies, VA of counting fingers, hand motion, light perception, and no light perception were converted to logMAR VA values of 1.85, 2.30, 2.80, and 2.90, respectively, for statistical analyses [14,15]. The difference between categorical and continuous variables was analyzed using the Mann–Whitney U test and chi-square test. Differences between variables before and after treatment were analyzed using a paired *t*-test.

## 3. Results

This study included 30 eyes of 30 patients. The mean age of the patients was 58.8 ± 16.7 years, and 25 eyes were of male patients (83.3%). Patients were divided into two groups according to the type of farming that contributed to the eye injury. Fourteen eyes were injured during farm work at a vineyard (vineyard group) and 16 were injured during other farm work (other farming groups). Table 1 shows the baseline demographics of patients in the two groups. Patients were younger and the proportion of intraocular foreign bodies was lower in the vineyard group than in the other farming group. The quarterly of the injury was also different between the two groups. In the vineyard group, 10 eyes (71.4%) presented to our hospital in Q1 (from January to March), whereas in the other farming group, 10 eyes (62.5%) presented in Q2–Q3 (from April to September, *p* = 0.013). None of the 30 eyes wore any eye protection. Figure 1 illustrates the representative case of vineyard-associated eye injury.

Among the eye injuries, corneal penetration was observed in 17 eyes (56.7%), traumatic cataract in 10 (33.3%), retinal detachment in three (10.0%), intraocular foreign body in nine (30.0%), and infection in three (10.0%, Table 1). Table 2 shows the causes of eye injury in the two groups. Eye injury by wires was the dominant cause in the vineyard group, while flying objects during grass trimmer handling was the dominant cause in the other farming group. In the vineyard group, one eye showed endophthalmitis caused by the penetration of the wire and the pathogen was not detected. The patient was administered topical and general antibacterial agents as an empirical therapy and was successfully treated. In the other farming group, two eyes with fungal corneal ulcers were caused by being scratched by a branch and the fungi species were unknown. These two patients were treated with topical and general antifungal agents as empirical therapy and were cured successfully. During the follow-up period, no complication was detected except in one eye in the other farming group. In that eye, the corneal wound due to the flying metal object during handling the grass trimmer did not close at initial surgery and needed additional corneal suture.

On initial presentation, 25 of the 30 eyes required emergency ophthalmic surgery. Among the 25 eyes, nine underwent corneal suture only; seven underwent corneal suture and lensectomy; three underwent corneal suture, lensectomy, and pers plana vitrectomy; and two underwent reconstruction of the lacrimal canaliculi. Table 3 shows the breakdown of ophthalmic surgeries. Among the 25 eyes, four eyes (36.4%) in the vineyard group and four eyes (28.6%) in the other farming group underwent secondary surgery (*p* = 0.83). Most of these surgeries involved a secondary insertion of the intraocular lens for aphakic eyes after the initial lensectomy. There was no significant difference in the mean number of surgeries between the two groups (1.07 ± 0.73 in the vineyard group vs. 1.13 ± 0.62 in the other farming group, *p* = 0.83).

The mean (median) follow-up period was 14.4 ± 27.9 (3.5) months. At the final follow-up, the mean BCVA significantly improved to 0.30 ± 0.57 from 0.83 ± 0.94 at baseline (*p* = 5.8 × 10^−4^). Figure 2 illustrates the box-and-whisker plot of the mean and median logMAR BCVA at baseline and final follow-up. Although the mean improvement in logMAR BCVA in the vineyard group tended to be greater than that in the other farming group, this difference was not statistically significant (*p* = 0.079, Figure 2). Table 4 shows the differences in the distribution of the final BCVA between the present study and the OTS study in each OTS category. As a result, the distribution of the final BCVA in this study was significantly better than that in the OTS study, except for categories 1 and 5. We also compared the distribution of the final BCVA between the vineyard group and the other farming group in each OTS category (Table 5). As a result, the distribution of the final BCVA in the vineyard group was significantly better than that in the other farming group, except for categories 1 and 5. The difference in the mean OTS raw score sum was not significant between the two groups (84.2 ± 16.0 in the vineyard group vs. 91.8 ± 13.5 in the other farming group, *p* = 0.17).

## 4. Discussion

In this study, we retrospectively investigated the clinical course of farm work-associated eye injuries. To the best of our knowledge, this is the first study to investigate farm work-associated eye injuries in vineyards. The results showed that eye injuries at vineyards were dominantly caused by wire penetration compared to other farm work-associated eye injuries. We believe that this unique property of eye injury in vineyards was mostly caused by the use of grape shelves. The use of grape shelves (overhead trellis) is a widely adopted style of grape cultivation in Asian countries, including Japan [16]. Grape shelves are set up to fix grapevines and are usually built in the winter season with thick wires. Grape shelves were approximately 180 cm in height; thus, farmers were forced to set them in a labored posture (Figure 3). In this study, wire penetration occurred during the cutting or winding of grape shelves. Although eye injuries have an annual peak from spring to summer [17], vineyard-associated eye injuries in this study were mostly observed in Q1 (from January to March). We considered this to be associated with the operation of the grape shelves. In addition, the significant difference in the mean age of patients between the two groups was noteworthy. Grape cultivation is predominant in Yamanashi Prefecture; hence, farmers engaged in grape breeding are younger than those engaged in other farm work.

To investigate the visual prognosis after ocular trauma, Kuhn et al. advocated the OTS in their published research. Several reports have verified the prognostic value of OTS. Purtskhvanidze et al. reported the treatment outcomes of ocular injuries owing to rotating wire brushes and concluded that OTS provided prognostic information for these patients [18]. In the previous study, only 5% of the patients were categorized as OTS 1 and 2 and 90% were categorized as OTS 3, 4, or 5. The visual prognosis was also relatively favorable, reflecting the OTS distribution. Yaşa et al. recently used OTS for open-globe injuries associated with metallic foreign bodies and reported that the final VA of the patients showed visual outcomes equivalent to those estimated using OTS [19]. In contrast, the patients in this study had better visual outcomes than those estimated using the OTS, except for categories 1 and 5. We attribute this discrepancy between the present study and previous studies to the difference in sample size and the relatively small entry wounds, including the penetration of wires and intraocular foreign bodies in this study. Another reason for the relatively preferable visual outcome of eye injuries in a vineyard was that corneal penetration by wires was the primary cause of injury and that wires barely reached the neurosensory retina. Furthermore, the fact that there were no globe ruptures or perforating injuries in this study could have also resulted in favorable visual outcomes.

In agricultural situations, grass trimmers have the potential to cause open-globe injuries [20]. Recently, Supreeyathitikul et al. reported a long-term observational study investigating the epidemiology of open-globe injury in a tertiary referral center around the agricultural region [8]. In the retrospective study, they reported that as much as 24% of open-globe injuries were caused by flying objects from electric grass trimmers. Similar to their study, our results showed that eye injuries in the other farming group were predominantly caused by handling grass trimmers.

Several reports have emphasized the necessity of wearing eye protection to prevent the occupational risk of eye injuries [21,22]. Chatterjee et al. investigated the effectiveness of safety eyewear in preventing eye injury in agricultural populations [23]. They reported that farmers harvesting paddy with safety eyewear had a 94% lower risk of ocular trauma than those without any safety eyewear. In the present study, none of the participants wore any safety eyewear, resulting in ocular injuries. Given that the participants included in this study presented to the tertiary referral suite of our hospital from the ophthalmology clinic, it is suggested that the real-life usage rate of eyewear among all farmers in Japan may not be reflected in this study. However, considering that the occupational risk of ocular injury was preventable with appropriate safety eyewear in the previous studies [24,25,26], most of the eye injuries included in this study might have been preventable with safety eyewear.

Our study has some limitations. The first limitation is the small number of patients. In this study, we included 30 eyes over 13 years. Given that the number of the farm household in Yamanashi Prefecture is 30 to 40 thousand, the prevalence of farm work-associated eye injuries in this study (estimated at 0.6–0.7/1000 worker years) was less than that reported in the previous studies (11.3/1000 to 8.7/10,000 worker years) [4,27,28]. Thus, the prevalence of farm work-associated eye injury in this study might be underestimated and have affected the results. The second limitation is the retrospective nature of the study design. A further prospective investigation of the vineyard-associated eye injury with a large number of patients is expected.

In conclusion, farm work-associated eye injuries in vineyards have characteristic properties compared to those during other farm work. Corneal penetration by wires was the major cause of injuries, particularly during the winter period. Therefore, ophthalmologists should be aware of farmers’ risk of ocular injury and alert them to use safety eyewear to prevent eye injury.

## Figures and Tables

**Figure 1 jcm-11-07079-f001:**
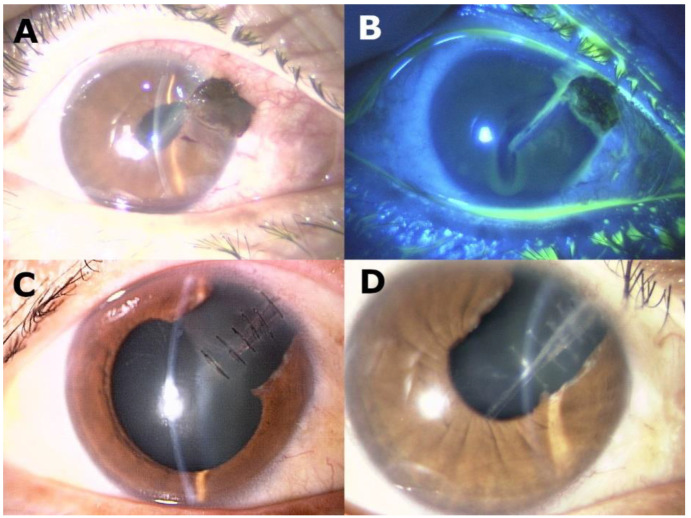
A representative case of a 54-year-old man with right corneal penetration by wires during farming at a vineyard. (**A**) Slit lamp biomicroscopy of the right eye at baseline reveals corneal penetration in the 2:00 direction accompanied by iris prolapse. The right BCVA at baseline was hand motion. (**B**) Slit lamp biomicroscopy of the right eye after fluorescein staining at baseline reveals a corneal wound from the 2:00 direction toward the corneal center. (**C**) Slit lamp biomicroscopy of the right eye at 1 month from baseline shows a completely sutured corneal wound with a partial defect of the iris in the 2:00 direction. (**D**) Slit lamp biomicroscopy of the right eye at 3 months from baseline reveals complete closure of the corneal wound after suture removal. His right logMAR BCVA improved to 0.05.

**Figure 2 jcm-11-07079-f002:**
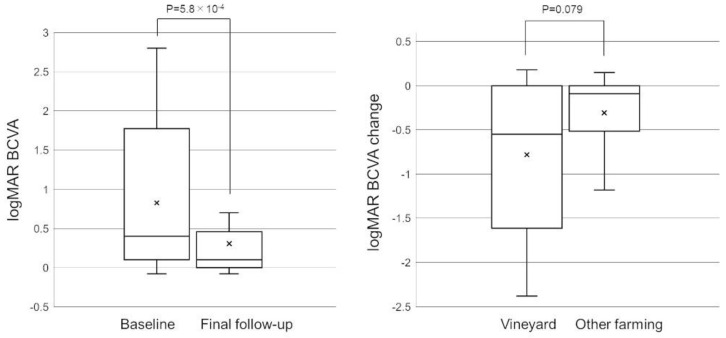
Box-and-whisker plot of mean and median logMAR BCVA changes. The ‘×’ marks in the boxes indicate the mean BCVA/BCVA changes. The horizontal lines in the boxes indicate median BCVA/BCVA changes, and the top/bottom edges of the boxes indicate 25/75 percentile. The top/bottom edges of the whiskers indicate maximum/minimum values. (**Left**) Box-and-whisker plots of mean and median logMAR BCVA at baseline and the final follow-up. At the final follow-up, mean BCVA significantly improved to 0.30 ± 0.57 from 0.83 ± 0.94 at baseline (*p* = 5.8 × 10^−4^). (**Right**) Box-and-whisker plots of mean and median logMAR BCVA changes in the vineyard and other farming groups. There was no significant difference in the BCVA improvement between the two groups (*p* = 0.079).

**Figure 3 jcm-11-07079-f003:**
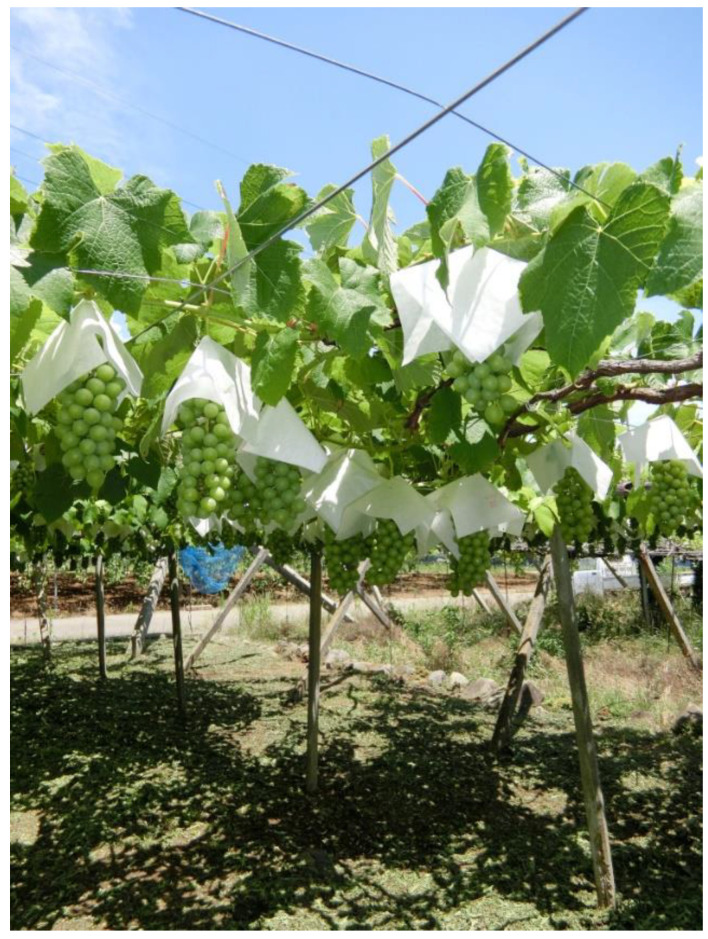
Picture of grape shelves in Kofu basin. A common overhead trellis consists of support pillars and thick wires.

**Table 1 jcm-11-07079-t001:** Baseline characteristics of the subject.

	All Subjects *n* = 30	Vineyard-Associated Injury *n* = 14	Other Farmwork-Associated Injury *n* = 16	*p* Value (Vineyard vs. Other Farmwork)
age	58.8 ± 16.7	51.3 ± 18.8	65.4 ± 11.6	0.018
sex male	25 (83.3%)	12 (85.7%)	13 (81.3%)	0.74
baseline BCVA	0.83 ± 0.94	1.06 ± 1.03	0.64 ± 0.84	0.22
time from injury (days)	0.50 ± 0.90	0.21 ± 0.58	0.75 ± 1.1	0.10
Open globe injury (%)	20 (66.7%)	9 (64.2%)	11 (68.8%)	0.79
Corneal penetration (%)	17 (56.7%)	6 (42.9%)	11 (68.8%)	0.51
Traumatic cataract (%)	10 (33.3%)	3 (21.4%)	7 (43.8%)	0.19
Intraocular foreign body (%)	9 (30.0%)	0 (0%)	9 (56.3%)	8.0 × 10^−4^
Retinal detachment (%)	3 (10.0%)	2 (14.3%)	1 (6.3%)	0.46
Infection (%)	3 (10.0%)	1 (7.1%)	2 (12.5%)	0.63
Wearing eye protect	0	0	0	NA
Date of injury (%)				
Q1	12 (40.0%)	10 (71.4%)	2 (12.5%)	0.013
Q2	5 (16.7%)	1 (7.1%)	4 (25.0%)	
Q3	8 (26.7%)	2 (14.3%)	6 (37.5%)	
Q4	5 (16.7%)	1 (7.1%)	4 (25.0%)	

BCVA: best-corrected visual acuity, NA: not applicable, Q1: from January to March, Q2: from April to June, Q3: from July to September, Q4: from October to December.

**Table 2 jcm-11-07079-t002:** Causes of eye injuries in the two groups.

	All Subjects *n* = 30	Vineyard-Associated Injury *n* = 14	Other Farm Work-Associated Injury *n* = 16	*p*-Value (Vineyard vs. Other Farm Work)
Injury by wires	11 (36.7%)	11 (78.6%)	0	1.0 × 10^−5^
Frying object during handling grass trimmer	11 (36.7%)	0	11 (68.8%)	1.0 × 10^−4^
Ocular contusion by branches	5 (16.7%)	1 (7.1%)	4 (25.0%)	0.19
Ocular contusion by pipes	2 (6.7%)	1 (7.1%)	1 (6.3%)	0.92
Others	1 (3.3%)	1 (7.1%)	0	0.28

**Table 3 jcm-11-07079-t003:** Breakdown of the emergency ophthalmic surgeries.

	All Subjects *n* = 25	Vineyard-Associated Injury *n* = 11	Other Farmwork-Associated Injury *n* = 14	*p*-Value (Vineyard vs. Other Farmwork)
Corneal suture only	9 (36.0%)	3 (27.2%)	6 (42.9%)	0.42
Corneal suture and lensectomy	7 (28.0%)	1 (9.1%)	6 (42.9%)	0.06
Corneal suture, lensectomy and vitrectomy	3 (12.0%)	2 (18.2%)	1 (7.1%)	0.40
Vitrectomy only	2 (8.0%)	2 (18.2%)	0	0.10
Reconstruction of lacrimal canaliculi	2 (8.0%)	1 (9.1%)	1 (7.1%)	0.86
Scleral and/or conjunctival suture	2 (8.0%)	2 (18.2%)	0	0.10

**Table 4 jcm-11-07079-t004:** Comparison of final best-corrected visual acuities and OTS distribution between OTS study and this study.

OTS Category	Final BCVA					*p* Value
	NLP A/B	LP/HM A/B	1/200–19/200 A/B	20/200–20/50 A/B	>20/40 A/B	
1	0/74	0/15	0/7	0/3	0/1	1.00
2	0/27	33/26	33/18	0/15	33/15	<1.0 × 10^−5^
3	0/2	0/11	0/15	57/31	43/41	<1.0 × 10^−5^
4	0/1	0/2	0/3	40/22	60/73	3.9 × 10^−3^
5	0/0	0/1	0/1	0/5	100/94	0.13

OTS: Ocular Trauma Score, BCVA: best-corrected visual acuity, NLP: No light perception, LP: Light perception, HM: hand motion, A: Our study results (%), B: OTS study results (%). Chi-square test.

**Table 5 jcm-11-07079-t005:** Comparison of final best-corrected visual acuities and OTS distribution between vineyard group and other farming group.

OTS Category	Final BCVA					*p* Value
	NLP A/B	LP/HM A/B	1/200–19/200 A/B	20/200–20/50 A/B	>20/40 A/B	
1	0/0	0/0	0/0	0/0	0/0	1.00
2	0/0	0/100	50/0	0/0	50/0	<1.0 × 10^−5^
3	0/0	0/0	0/0	40/100	60/0	<1.0 × 10^−5^
4	0/0	0/0	0/0	0/50	100/50	<1.0 × 10^−5^
5	0/0	0/0	0/0	0/0	100/100	1.00

OTS: Ocular Trauma Score, BCVA: best-corrected visual acuity, NLP: No light perception, LP: Light perception, HM: hand motion, A: Our study results (%), B: OTS study results (%). Chi-square test.

## Data Availability

The data presented in this study are available on request from the corresponding author. The data are not publicly available due to privacy restrictions.

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
