# Peer review of "Characteristics of Grape Shelf Eye Injuries at Vineyards in Japan"

_jcm, 2022, doi:10.3390/jcm11237079_

Round 1

Reviewer 1 Report

The authors retrospectively investigated the characteristics and visual outcomes of farm work-associated eye injuries at vineyards. The main body of the paper is well written, however I would recommend moderate English editing and there are some minor suggestions that should be considered:

1)     In table please change “gender” to “sex”

2)     Line 99 – using hand motion is not LogMAR scale – consider to use hand motion or LogMAR scale

3)     Please add additional information how the presumed infections were treated (endophthalmitis  - vitrectomy? and fungal keratitis) and as the cultures were negative, probably an empirical therapy was started? Was the treatment successful? Were PCR methods also used and negative?

4)     Please add information how the patients are medically treated after the emergency surgery (please add information on topical and /or systemic therapy).

5)     Do you ask for tetanus vaccination?

Reviewer 2 Report

This is a truly valuable paper warning potential danger of the vineyard works in Japan.

It is recommended that title should be slightly modified, which include Japanese vineyards, since overhead trellis is only adopted in Japan or Asia as authors mentioned.

Reviewer 3 Report

Farm work-associated eye injuries in vineyards have characteristic properties compared to those during other farm work. Corneal penetration by wires was the major cause of injuries, particularly during the winter period. Therefore, ophthalmologists should be aware of farmers’risk of ocular injury and alert them to use safety eye wear to prevent eye injury.

The complication of all these cases was not mentioned, such as intraocular infection or tramatic uveitis and others. would you give detailed complications of these cases?

Reviewer 4 Report

In this paper, the authors described characteristics and visual outcomes of farm work-associated eye injuries at vineyards in Japan, and compared them with other farming caused eye injuries. The study included 30 farm work-associated eye injuries during 13 years, all from Yamanashi Prefecture, Japan. The small sample size and single center retrospectively reviewed design limited the study’s clinical significance and research value. And the manuscript will not fit well with the scope of the Special Issue "Corneal Surgery: From Innovation to Clinical Praxis".

 1. The authors compared the differences of OTS scores between the present study and OTS study. But these two studies have big difference in sample size and causes of injuries. It will be more research significance to compare OTS scores between vineyard group and other farming group.

2. In the present study, none of the participants wore any safety eyewear. The data to support the conclusion that “most of the eye injuries included in this study might have been preventable with safety eyewear” was not insufficient.

Round 2

Reviewer 4 Report

In their revised manuscript, the authors have responded to the comments provided in the original one. They also included more discussions for helping comprehensively explain the results. But there are still several issues need to be addressed.

The authors mentioned “The ocular trauma score (OTS) of the eyes included in this study was calculated based on initial BCVA and the type of injury.” But in Table 4 and 5, they all showed as final BCVA. In OTS study, the scores were from initial visual acuity. Final BCVA is not only related to trauma, but also affected by the treatment process to a large extent. The authors should make this more clear.

 All tables should be showed as uniformly three-line tables, and displayed on one page.
